# Something for (almost) nothing: improving deep ensemble calibration using unlabeled data

**Konstantinos Pitas**
Statify Team
Inria Grenoble Rhône-Alpes
`pitas.konstantinos@inria.fr`

**Julyan Arbel**
Statify Team
Inria Grenoble Rhône-Alpes
`julyan.arbel@inria.fr`

## Abstract

We present a method to improve the calibration of deep ensembles in the small training data regime in the presence of unlabeled data. Our approach is extremely simple to implement: given an unlabeled set, for each unlabeled data point, we simply fit a different randomly selected label with each ensemble member. We provide a theoretical analysis based on a PAC-Bayes bound which guarantees that if we fit such a labeling on unlabeled data, and the true labels on the training data, we obtain low negative log-likelihood and high ensemble diversity on testing samples. Crucially, each ensemble member can be trained independently from the rest (apart from the final validation/test step) making a parallel or distributed implementation extremely easy.

## 1 Introduction

Deep ensembles have gained widespread popularity for enhancing both the testing accuracy and calibration of deep neural networks. Importantly, deep ensembles outperform Bayesian approaches for the same number of posterior samples (Arbel et al., 2023). Both empirically and theoretically, the performance of deep ensembles is intrinsically tied to their diversity (Fort et al., 2019; Masegosa, 2020). By averaging predictions from a more diverse set of models, we mitigate prediction bias and thereby enhance overall performance.

The conventional approach to introducing diversity within deep ensembles involves employing distinct random initializations for each ensemble member (Lakshminarayanan et al., 2017). As a result, these ensemble members converge towards different modes of the loss landscape, each corresponding to a unique predictive function. This baseline technique is quite difficult to surpass. Nevertheless, numerous efforts have been made to further improve deep ensembles by explicitly encouraging diversity in their predictions (Ramé & Cord, 2021; Yashima et al., 2022; Masegosa, 2020; Matteo et al., 2023).

These approaches typically encounter several challenges, which can be summarized as follows: *The improvements in test metrics tend to be modest, while the associated extra costs are substantial.* Firstly, diversity-promoting algorithms often involve considerably more intricate implementation details compared to randomized initializations. Secondly, the computational and memory demands of existing methods exceed those of the baseline by a significant margin. Additionally, some approaches necessitate extensive hyperparameter tuning, further compounding computational costs.

In light of these considerations, we introduce $\nu$-ensembles, an algorithm designed to improve deep ensemble calibration and diversity with minimal deviations from the standard deep ensemble workflow. Moreover, our algorithm maintains the same computational and memory requirements as standard

Workshop on Advancing Neural Network Training at 37th Conference on Neural Information Processing Systems (WANT@NeurIPS 2023).

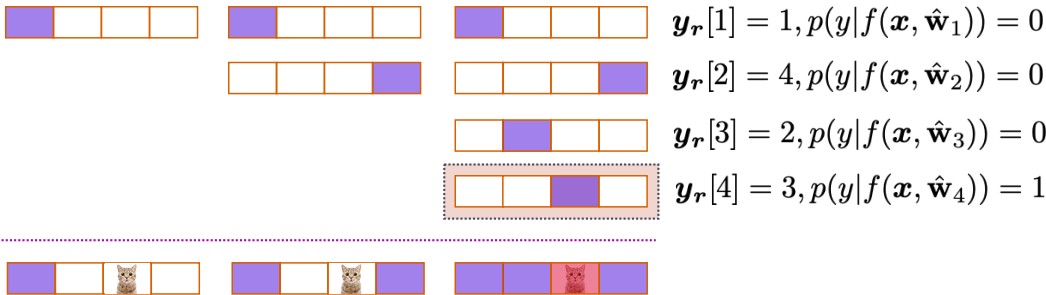

Figure 1: **Motivating $\nu$-ensembles.** Consider a 4-class classification problem and an unlabeled sample $\boldsymbol{x}$ with true label $y$. We sample $K = 4$ labels without replacement $\boldsymbol{y_r} = [1, 4, 2, 3]$ and fit them perfectly with ensemble members $\{\hat{\mathbf{w}}_1, \hat{\mathbf{w}}_2, \hat{\mathbf{w}}_3, \hat{\mathbf{w}}_4\}$. As we have sampled exhaustively all classes for this classification problem, exactly one of the sampled labels will be the correct one. The corresponding ensemble member $\hat{\mathbf{w}}_4$ will learn a useful feature from the input label pair $(\boldsymbol{x}, y)$. Noting that $p(y|\boldsymbol{x}, \hat{\mathbf{w}}_i)$ *is with respect to the true label* $y$, $p(y|\boldsymbol{x}, \hat{\mathbf{w}}_1) = 0, p(y|\boldsymbol{x}, \hat{\mathbf{w}}_2) = 0, p(y|\boldsymbol{x}, \hat{\mathbf{w}}_3) = 0, p(y|\boldsymbol{x}, \hat{\mathbf{w}}_4) = 1$ and the empirical variance will be $\hat{\mathbf{V}}(\hat{\rho}) = \frac{1}{2}\left[\frac{1}{K}\sum_j\left[(p(y|\boldsymbol{x}, \mathbf{w}_j) - \frac{1}{K}\sum_i(p(y|\boldsymbol{x}, \mathbf{w}_i)))^2\right]\right] = \frac{1}{2}\frac{4-1}{4}\cdot\frac{1}{4} = \frac{1}{2}\cdot\frac{3}{16}$ as computed in Proposition 1.

deep ensembles, resulting in linear increases in computational costs with the size of the unlabeled dataset.

**Our contributions**

- Given an ensemble of size $K$ and an unlabeled set, we propose an algorithm that generates for each unlabeled data point $K$ random labels without replacement and assigns from these a single random label to each ensemble member. For each ensemble member we then simply fit the training data (with its true labels) as well as the unlabeled data (with the generated random labels). See Figure 1. Each ensemble member can be trained independently facilitating a parallel or distributed implementation.

- We provide a PAC-Bayesian analysis of the test performance of the trained ensemble in terms of negative log-likelihood and diversity. On average, the final ensemble is guaranteed to be diverse, accurate, and well-calibrated on test data.

- We provide experiments for the in-distribution setting that demonstrate that for small to medium-sized training sets, $\nu$-ensembles are better calibrated than standard ensembles in the most common calibration metrics.

- We also provide detailed experiments in the out-of-distribution setting and demonstrate that $\nu$-ensembles remain significantly better calibrated than standard ensembles for a range of common distribution shifts.

## 2 Small to medium-sized training set setting

In the laboratory setting, deep learning models are typically trained and evaluated using large highly curated, and labeled datasets. However, real-world settings usually differ significantly. Labeled datasets are often small as the acquisition and labeling of new data is expensive, time-consuming, or simply not feasible. A small labeled training set is also often accompanied by a larger unlabeled set.

The small data regime has been explored in a number of works (Ratner et al., 2017; Balestriero et al., 2022; Zoph et al., 2020; Sorscher et al., 2022; Bornschein et al., 2020; Cubuk et al., 2020; Fabian et al., 2021; Zhao et al., 2019; Foong et al., 2021; Perez-Ortiz et al., 2021), both theoretical and practical. Two of the most common approaches for dealing with few training data, are using an ensemble of predictors, and/or using data augmentation to artificially create a larger training set.

We test our proposed $\nu$-ensembles for a range of training set sizes, while applying data augmentation, and have found that we get performance gains for small to medium-sized training sets (1K - 10K samples). We emphasize that the "small data" regime is relative; more complex distributions require more data. As such $\nu$-ensembles can be effective beyond these thresholds.

## 3    Related work on improvements of deep ensembles

A number of approaches have been proposed to improve upon standard deep ensembles.

**Diversity promoting objectives.** Ramé & Cord (2021) propose to use a discriminator that forces the latent representations of each ensemble member just before the final classification layer to be diverse. They show consistent improvements for large-scale settings in terms of test accuracy and other metrics, however, their approach requires very extensive hyperparameter tuning. Yashima et al. (2022) encourage the latent representations just before the classification layer to be diverse by leveraging Stein Variational Gradient Descent (SVGD). They show improvements in robustness to non-adversarial noise. However, they do not show improvements over Ramé & Cord (2021) in other metrics.

Masegosa (2020); Ortega et al. (2022) propose optimizing a second-order PAC-Bayes bound to enforce diversity. In practice, this means estimating the mean likelihood of a true label across different ensemble members and "pushing" the different members to estimate a different value for their own likelihood. The authors show improvements for small-scale experiments, however, this comes at the cost of two gradient evaluations per data sample at each optimization iteration. The method closest to our approach is the very recently proposed Agree to Disagree algorithm (Matteo et al., 2023). Agree to disagree forces ensemble members to disagree with the other members on unlabeled data. Crucially, however, (and in contrast to our approach) the ensemble is constructed greedily, where a single new member is added at a time and is forced to disagree with the previous ones. The method is also evaluated only in the OOD setting.

The above methods exhibit all the shortcomings we previously described, where the cost of implementation, tuning and training cannot easily be justified: 1) the implementation differs significantly from standard ensembles (Ramé & Cord, 2021; Yashima et al., 2022; Masegosa, 2020; Matteo et al., 2023); 2) the computational complexity increases significantly (Ramé & Cord, 2021; Matteo et al., 2023); 3) and the algorithm requires extensive hyperparameter tuning (Ramé & Cord, 2021).

**Bayesian approaches.** One can also approach ensembles as performing approximate Bayesian inference (Wilson & Izmailov, 2020). Under this view, a number of approaches that perform approximate Bayesian inference can also be seen as constructing a deep ensemble (Izmailov et al., 2021; Wenzel et al., 2020a; Zhang et al., 2020; Immer et al., 2021; Daxberger et al., 2021). The samples from the approximate posterior that form the ensemble can be sampled locally around a single mode using the Laplace approximation (Immer et al., 2021; Daxberger et al., 2021) or from multiple modes using MCMC (Izmailov et al., 2021; Wenzel et al., 2020a; Zhang et al., 2020). While some approaches resort to stochastic MCMC approaches for computational efficiency (Wenzel et al., 2020a; Zhang et al., 2020), the authors of Izmailov et al. (2021) apply full-batch Hamiltonian Monte Carlo which is considered the gold standard in approximate Bayesian inference. D'Angelo & Fortuin (2021) propose a repulsive approach in terms of the neural network weights. They show that the resulting ensemble can be seen as Bayesian, however, they do not demonstrate consistent improvements across experimental setups.

One would hope that the regularizing effect of the Bayesian inference procedure would improve the resulting ensembles. Unfortunately, approximate Bayesian inference approaches are typically outperformed by *standard* deep ensembles (Ashukha et al., 2019). In particular, to achieve the same misclassification or negative log-likelihood error, MCMC approaches typically require many more ensemble members than standard ensembles.

**Complementary works.** Some works on diverse ensembles are compatible with our approach and can be used in conjunction with it.

Wenzel et al. (2020b) propose to induce diversity by training on different random initializations as well as different choices of hyperparameters such as the learning rate and the dropout rates in different layers. Ensemble members can be trained independently, and the approach results in consistent gains over standard ensembles. As we also train each ensemble member independently we could use

hyperparameter ensembling to improve diversity. Jain et al. (2022) propose to create different training sets for each ensemble member using image transformations (for example edge detection filters) to bias different ensemble members towards different features. In a similar vein, Loh et al. (2023) encourage different ensemble members to be invariant or equivariant to different data transformations. These approaches can also be used in conjunction with our method to further increase diversity.

**Self-training.** Jain et al. (2022) propose to pseudo-label unlabeled data using deep ensembles trained on labeled data. These pseudo-labeled data are then used to retrain the ensemble. This approach (known as self-training, see Lee et al., 2013) can improve significantly standard ensembles. We note however that it is complicated to implement and costly. First, unlabeled data have to be labeled in multiple rounds, a fraction at a time. Also, to be fully effective, ensembles have to be "distilled" into a final single network. Finally, care has to be taken that ensemble members capture diverse features. By contrast, our method requires a single random labeling of unlabeled data, followed by standard training and introduces a single hyperparameter that is easy to tune.

## 4 Diversity through unlabeled data

We now introduce some notation and then make precise our notions of train and test performance, as well as diversity.

We denote the learning sample $(X, Y) = \{(\boldsymbol{x}_i, y_i)\}_{i=1}^n \in (\mathcal{X} \times \mathcal{Y})^n$, that contains $n$ input-output pairs, and use the generic notation $Z$ for an input-output pair $(X, Y)$. Observations $(X, Y)$ are assumed to be sampled randomly from a distribution $\mathcal{D}$. Thus, we denote $(X, Y) \sim \mathcal{D}^n$ the i.i.d observation of $n$ elements. We consider loss functions $\ell : \mathcal{F} \times \mathcal{X} \times \mathcal{Y} \to \mathbb{R}$, where $\mathcal{F}$ is a set of predictors $f : \mathcal{X} \to \mathcal{Y}$. We also denote the empirical risk $\hat{\mathcal{L}}^\ell_{X,Y}(f) = (1/n)\sum_i \ell(f, \boldsymbol{x}_i, y_i)$. We denote $\ell_{\mathrm{nll}}(f, \boldsymbol{x}, y) = -\log(p(y|\boldsymbol{x}, f))$ the negative log-likelihood, where we assume that the outputs of $f$ are normalized to form a probability distribution, and $p(y|\boldsymbol{x}, f)$ the probability of label $y$ given $\boldsymbol{x}$ and $f$.

Now let us assume that $f$ is a deep neural network architecture, and $\hat{\rho}(\mathbf{w}) = \frac{1}{K}\sum_i \delta(\mathbf{w} = \hat{\mathbf{w}}_i)$ is a set of minima that form a deep ensemble. We are typically interested in minimizing $\mathbf{E}_{(y,\boldsymbol{x})\sim\mathcal{D}}\left[-\ln\frac{1}{K}\sum_i [p(y|\boldsymbol{x}, f(\boldsymbol{x}; \hat{\mathbf{w}}_i))]\right]$, the loss over new samples drawn from $\mathcal{D}$ for the ensemble predictor, that is: a predictor where we average the probabilities estimated per class by each ensemble member $\frac{1}{K}\sum_i p(y|\boldsymbol{x}, f(\boldsymbol{x}; \hat{\mathbf{w}}_i))$. The standard deep ensemble algorithm then simply minimizes $\forall i, \min_{\mathbf{w}_i} \hat{\mathcal{L}}^{\ell_{\mathrm{nll}}}_Z(f(\boldsymbol{x}; \hat{\mathbf{w}}_i))$ for some training set $Z$.

Let us now assume that we have access not only to a training set $Z$ but also to an unlabeled set $U$ of size $m$. We can then present a PAC-Bayes bound* that links the loss on new test data to the loss on the training data as well as the diversity of the ensemble predictions on the unlabeled data.

**Theorem 1.** *With high probability over the training set $Z$ and the unlabeled set $U$ drawn from $\mathcal{D}$, for an ensemble $\hat{\rho}(\mathbf{w}) = \frac{1}{K}\sum_i \delta(\mathbf{w} = \hat{\mathbf{w}}_i)$ on $\mathcal{F}$ and all $\gamma \in (0, 2)$ simultaneously*

$$\mathbf{E}_{(y,\boldsymbol{x})\sim\mathcal{D}}\left[-\ln\frac{1}{K}\sum_i [p(y|\boldsymbol{x}, f(\boldsymbol{x}; \hat{\mathbf{w}}_i))]\right]$$
$$\leq \frac{1}{K}\sum_i \left[\hat{\mathcal{L}}^{\ell_{\mathrm{nll}}}_Z(f(\boldsymbol{x}; \hat{\mathbf{w}}_i))\right] - \left(1 - \frac{\gamma}{2}\right)\hat{\mathbf{V}}(\hat{\rho}) + \frac{1}{K}\sum_i h\left(\|\hat{\mathbf{w}}_i\|_2^2\right), \quad (1)$$

*where*

$$\hat{\mathbf{V}}(\hat{\rho}) = \frac{1}{2m}\sum_U \left[\frac{1}{K}\sum_j \left[\left(p(y|\boldsymbol{x}, f(\boldsymbol{x}, \hat{\mathbf{w}}_j)) - \frac{1}{K}\sum_i p(y|\boldsymbol{x}, f(\boldsymbol{x}, \hat{\mathbf{w}}_i))\right)^2\right]\right] \quad (2)$$

*is the empirical variance of the ensemble, and $h : \mathbb{R}^+ \to \mathbb{R}^+$ is a strictly increasing function.*

The term $\frac{1}{K}\sum_i \left[\hat{\mathcal{L}}^{\ell_{\mathrm{nll}}}_Z(f(\boldsymbol{x}; \hat{\mathbf{w}}_i))\right]$ is simply the average negative log-likelihood of all the ensemble members on the training set $Z$. The term $\hat{\mathbf{V}}(\hat{\rho})$ captures our notion of diversity for the deep ensemble.

---

*Variants of this bound have appeared in recent works for majority vote classifiers (Thiemann et al., 2017; Wu & Seldin, 2022; Masegosa et al., 2020; Masegosa, 2020). However, to the best of our knowledge, this particular version is novel in the deep ensemble case.

Specifically, given a sample $(\boldsymbol{x}, y)$ it is the empirical variance of the likelihood $p(y|\boldsymbol{x}, f)$ of the correct class $y$ over all the ensemble members. The terms $h\left(\|\hat{\mathbf{w}}_i\|_2^2\right)$ capture a notion of complexity of the deep ensemble. If this term is too large, then it is possible that the ensemble has memorized the training and unlabeled sets leading to poor generalization on new data. From the above, we see that for a deep ensemble to generalize well to new data one needs to minimize its average training error, while maximizing its variance.

One could attempt to optimize the RHS of (1) directly by setting $U = Z$, through gradient descent. However, this introduces unnecessary complexity to the optimization objective, necessitates that all ensemble members are trained jointly, and also neglects potentially useful unlabeled data. We thus crucially evaluate the variance on a new unlabeled set $U$ and not the training set $Z$. However, a careful reader would note that it is no longer possible to apply gradient descent directly to (1) as $\hat{\mathbf{V}}(\hat{\rho})$ depends on the unknown true label $y$. *We thus show in the following proposition that it is actually not necessary to know the true label $y$.* For each unlabeled sample $\boldsymbol{x}$, it simply suffices to draw $K$ labels randomly without replacement and assign each of them to a different member of the deep ensemble. Then for $K = c$ exactly one of these labels will be the correct one. If each ensemble member fits these random labels perfectly then we can compute the variance term analytically for $K \leq c$.

**Proposition 1.** *Assume an unlabeled set $U \in \mathcal{D}^m$, $c$ number of classes, and a labeling distribution $\mathcal{R}$ which for each sample $(\boldsymbol{x}, \cdot) \in U$ selects $K \leq c$ labels from $[1, \ldots, c]$ randomly without replacement such that $\boldsymbol{y_r} \in [1, \ldots, c]^K$. Let $\mathcal{A}$ be an algorithm that takes $\boldsymbol{y_r}$ as input and generates an ensemble $\hat{\rho}(\mathbf{w}) = \frac{1}{K} \sum_i \delta(\mathbf{w} = \hat{\mathbf{w}}_i)$ such that $\forall i, f(\boldsymbol{x}, \hat{\mathbf{w}}_i)$ perfectly fits $\boldsymbol{y_r}[i]$*

$$\mathbf{E}_{\hat{\rho} \sim \mathcal{A}}\left[\hat{\mathbf{V}}(\hat{\rho})\right] = \frac{K-1}{2cK} \tag{3}$$

*where the randomness is over $\boldsymbol{y_r}$ and we suppress the index for the different unlabeled points.*

*Proof.* The expectation of the variance term can be simply obtained by separating the cases when $y$ is and is not in the random labels $\boldsymbol{y_r}$ as follows

$$\mathbf{E}_{\hat{\rho} \sim \mathcal{A}}\left[\hat{\mathbf{V}}(\hat{\rho})\right] = \mathbf{E}_{\hat{\rho} \sim \mathcal{A}}\left[\frac{1}{2m} \sum_U \left[\frac{1}{K} \sum_j \left[\left(p(y|\boldsymbol{x}, \mathbf{w}_j) - \frac{1}{K} \sum_i (p(y|\boldsymbol{x}, \mathbf{w}_i))\right)^2\right]\right]\right]$$

$$= \frac{1}{2m} \sum_U \left[\frac{1}{K} \sum_j \left[\left(p(y|\boldsymbol{x}, \mathbf{w}_j) - \frac{1}{K} \sum_i (p(y|\boldsymbol{x}, \mathbf{w}_i))\right)^2\right] \cdot \int \mathbb{I}\{y \text{ in randomized labels}\} dr\right.$$

$$\left. + \frac{1}{K} \sum_j \left[\left(p(y|\boldsymbol{x}, \mathbf{w}_j) - \frac{1}{K} \sum_i (p(y|\boldsymbol{x}, \mathbf{w}_i))\right)^2\right] \cdot \int \mathbb{I}\{y \text{ not in randomized labels}\} dr\right]$$

$$= \frac{1}{2m} \sum_U \left[\frac{1}{K} \sum_j \left[\left(p(y|\boldsymbol{x}, \mathbf{w}_j) - \frac{1}{K} \sum_i (p(y|\boldsymbol{x}, \mathbf{w}_i))\right)^2\right] \cdot \int \mathbb{I}\{y \text{ in randomized labels}\} dr\right.$$

$$\left. + 0 \cdot \int \mathbb{I}\{y \text{ not in randomized labels}\} dr\right]$$

$$= \frac{1}{2m} \sum_U \left[\frac{1}{K} \sum_j \left[\left(p(y|\boldsymbol{x}, \mathbf{w}_j) - \frac{1}{K} \sum_i (p(y|\boldsymbol{x}, \mathbf{w}_i))\right)^2\right] \cdot \frac{K}{c}\right]$$

$$= \frac{1}{2mc} \sum_U \left[\sum_j \left[\left(p(y|\boldsymbol{x}, \mathbf{w}_j) - \frac{1}{K}\right)^2\right]\right]$$

$$= \frac{1}{2mc} \sum_U \left[\left(1 - \frac{1}{K}\right)^2 + \left(0 - \frac{1}{K}\right)^2 \cdot (K-1)\right]$$

$$= \frac{1}{2mc} \sum_U \left[\frac{K-1}{K}\right] = \frac{K-1}{2cK}.$$

$$\tag{4}$$

**Algorithm 1** $\nu$-ensembles

---

**Input:** Weight of the unlabeled loss $\beta$, $\ell_2$ regularization strength $\gamma$, training data $Z$, unlabeled data $U$, number of ensemble members $K$

**Output:** Ensemble $\mathcal{E}_K = \{\hat{\mathbf{w}}_1, \dots, \hat{\mathbf{w}}_K\}$

 1: **for** $i$ in $\{1, \dots, K\}$ **do**
 2:     $U_i \leftarrow \{\}$
 3:     **for** $\boldsymbol{x}$ in $U$ **do**
 4:         Sample $y$ randomly without replacement from $[1, \dots, c]$
 5:         $U_i \leftarrow U_i \cup (\boldsymbol{x}, y)$
 6:     **end for**
 7:     $\hat{\mathbf{w}}_i \leftarrow$ Random Initialization
 8:     $\min_{\hat{\mathbf{w}}_i} \hat{\mathcal{L}}_Z^{\ell_{\mathrm{nll}}}(f(\boldsymbol{x}; \hat{\mathbf{w}}_i)) + \beta \hat{\mathcal{L}}_{U_i}^{\ell_{\mathrm{nll}}}(f(\boldsymbol{x}; \hat{\mathbf{w}}_i)) + \gamma \|\hat{\mathbf{w}}_i\|_2^2$
 9: **end for**

---

$\square$

*Thus fitting $\boldsymbol{y_r} \sim \mathcal{R}$ guarantees in expectation through* (3) *a fixed level of variance, that strictly increases with the size of the ensemble.* Taking the expectation on both sides of (1) we can also derive a high probability bound on $\mathbf{E}_{\hat{\rho} \sim \mathcal{A}} \mathbf{E}_{(y, \boldsymbol{x}) \sim \mathcal{D}} \left[ -\ln \frac{1}{K} \sum_i [p(y|\boldsymbol{x}, f(\boldsymbol{x}; \hat{\mathbf{w}}_i))] \right]$ given multiple samples from $\hat{\rho} \sim \mathcal{A}$, and subject to additional conditions on the training set and complexity terms (namely boundedness). We defer the technical details to the Appendix.

We thus propose algorithm 1 to train $\nu$-ensembles. The proposed algorithm is extremely simple to implement. We simply need to construct $K$ randomly labeled sets $U_i$, such that all the sets $U_i$ contain different labels for all samples. We can then optimize

$$\hat{\mathcal{L}}_Z^{\ell_{\mathrm{nll}}}(f(\boldsymbol{x}; \hat{\mathbf{w}}_i)) + \beta \hat{\mathcal{L}}_{U_i}^{\ell_{\mathrm{nll}}}(f(\boldsymbol{x}; \hat{\mathbf{w}}_i)) + \gamma \|\hat{\mathbf{w}}_i\|_2^2 \tag{5}$$

with the optimization algorithm of our choice. In the above, $\beta$ is the weight placed on the randomly labeled samples. Notably, doing hyperparameter optimization over $\beta$ allows us to easily detect when $\nu$-ensembles improve upon standard ensembles using a validation set, as for $\beta = 0$ we recover standard ensembles. The term $\gamma \|\hat{\mathbf{w}}_i\|_2^2$ results from (1), and coincides we standard weight decay regularization. Crucially we rely on being able to fit random labels. We note that it is well known that deep neural networks can fit random labels perfectly (Zhang et al., 2021).

## 5 In-distribution and out-of-distribution experiments

We conducted two main types of experiments, evaluating (i) whether $\nu$-ensembles improve upon standard ensembles for in-distribution testing data, (ii) whether the gains of $\nu$-ensembles are robust to various distribution shifts.

To approximate the presence of unlabeled data using common classification datasets, given a training set $Z$, we reserve a validation set $Z_{\mathrm{val}}$, and a smaller training set $Z_{\mathrm{train}}$ and use the remaining datapoints as a pool for unlabeled data $U$. We keep the testing data $Z_{\mathrm{test}}$ unchanged.

### 5.1 In-distribution (ID) performance

To test in-distribution performance, we use the standard CIFAR-10 and CIFAR-100 datasets (Krizhevsky & Hinton, 2009). We explore a variety of dataset sizes. Specifically, for both datasets, we keep the original testing set such that $|Z_{\mathrm{test}}| = 10000$, and we use 5000 samples from the training set as unlabeled data $U$ and 5000 samples as validation data $Z_{\mathrm{val}}$. For training, we use datasets $Z_{\mathrm{train}}$ of size $1000, 2000, 4000, 10000$ and $40000$. We use three types of neural network architectures, a LeNet architecture LeCun et al. (1998), an MLP architecture with 2 hidden layers Goodfellow et al. (2016), and a WideResNet22 architecture Zagoruyko & Komodakis (2016). For both datasets, we used the standard augmentation setup of random flips + crops. We note that similar training-unlabeled set splits for CIFAR-10 and CIFAR-100 have been explored before in Alayrac et al. (2019); Jain et al. (2022).

We measure testing performance using accuracy as well as calibration on the testing set. Specifically, we measure calibration using the Expected Calibration Error (ECE) (Naeini et al., 2015), the

Table 1: **ID performance, 1000 training samples, 10 ensemble members.** $\nu$-ensembles retain approximately the same accuracy as standard ensembles. At the same time, they achieve significantly better calibration in all calibration metrics. The improvements are consistent across all tested architectures and both datasets. We also observe that the Mutual Information (MI) of $\nu$-ensembles is significantly lower than standard ensembles. Thus, $\nu$-ensembles are more diverse than standard ensembles, which explains their improved calibration. These empirical observations are also consistent with our theoretical analysis. Masegosa and Agree to Disagree ensembles typically undefit and have lower testing accuracy than both Standard and $\nu$-ensembles.

| Dataset / Aug | Method | Acc ↑ | ECE ↓ | TACE ↓ | Brier Rel. ↓ | NLL ↓ | MI ↓ |
|---|---|---|---|---|---|---|---|
| CIFAR-10 | Standard | 0.522 | 0.184 | 0.035 | 0.137 | 2.198 | 1.313 |
| / LeNet | Agree Dis. | 0.432 | 0.251 | 0.05 | 0.168 | 2.25 | 1.552 |
| | Masegosa | 0.492 | 0.103 | 0.024 | 0.073 | 1.454 | 1.179 |
| | $\nu$-ensembles | **0.5141** | 0.131 | 0.028 | 0.117 | 1.650 | 1.245 |
| CIFAR-10 | Standard | 0.398 | 0.238 | 0.05 | 0.162 | 2.197 | 1.615 |
| / MLP | Agree Dis. | 0.354 | 0.358 | 0.066 | 0.239 | 3.201 | 1.547 |
| | Masegosa | 0.383 | 0.024 | 0.024 | 0.068 | 1.768 | 1.711 |
| | $\nu$-ensembles | **0.401** | 0.098 | 0.023 | 0.092 | 1.767 | 1.559 |
| CIFAR-10 | Standard | 0.529 | 0.096 | 0.024 | 0.108 | 1.714 | 0.992 |
| / ResNet22 | Agree Dis. | 0.478 | 0.051 | 0.02 | 0.087 | 1.633 | 0.706 |
| | $\nu$-ensembles | **0.526** | 0.010 | 0.017 | 0.086 | 1.449 | 0.691 |
| CIFAR-100 | Standard | 0.151 | 0.301 | 0.007 | 0.216 | 9.434 | 2.228 |
| / LeNet | Agree Dis. | 0.113 | 0.229 | 0.007 | 0.156 | 7.568 | 1.628 |
| | Masegosa | 0.139 | 0.087 | 0.005 | 0.07 | 4.193 | 2.129 |
| | $\nu$-ensembles | **0.147** | 0.155 | 0.006 | 0.113 | 4.846 | 1.654 |
| CIFAR-100 | Standard | 0.102 | 0.253 | 0.007 | 0.16 | 5.926 | 3.093 |
| / MLP | Agree Dis. | 0.093 | 0.359 | 0.008 | 0.243 | 7.247 | 2.881 |
| | Masegosa | 0.093 | 0.257 | 0.008 | 0.16 | 6.134 | 3.103 |
| | $\nu$-ensembles | **0.103** | 0.04 | 0.004 | 0.049 | 4.171 | 2.807 |
| CIFAR-100 | Standard | 0.136 | 0.197 | 0.007 | 0.141 | 7.700 | 1.701 |
| | Agree Dis. | 0.132 | 0.172 | 0.007 | 0.124 | 6.831 | 1.708 |
| / ResNet22 | $\nu$-ensembles | **0.134** | 0.135 | 0.006 | 0.099 | 4.892 | 1.476 |

Thresholded Adaptive Calibration Error (TACE) (Nixon et al., 2019), the Brier Score Reliability (Brier Rel.) (Murphy, 1973), and the Negative Log-Likelihood (NLL). We also measure the diversity of the ensemble on the test set using the average mutual information between ensemble member predictions. More specifically for each ensemble we treat its output as a random variable giving values in $[1, \ldots, c]$. We compute the Mutual Information (MI) of this random variable between all ensemble pairs and take the average. Lower MI then corresponds to more diverse ensembles.

For both datasets, we first create an ensemble with $K = 10$ ensemble members and train each ensemble member using AdamW (Loshchilov & Hutter, 2017). For standard ensembles we simply minimize $\hat{\mathcal{L}}_Z^{\ell_{\mathrm{nll}}}(f(\boldsymbol{x}; \hat{\mathbf{w}}_i)) + \gamma \|\hat{\mathbf{w}}_i\|_2^2$ for each ensemble member using different random initializations. For $\nu$-ensembles we optimize (5). For hyperparameter tuning we perform a random search with 50 trials, using Hydra (Yadan, 2019). The details for the hyperparameter tuning ranges can be found in the Appendix. Table 1 presents the results for a training set of size 1000.

We see that $\nu$-ensembles have comparable accuracy to standard ensembles but with significantly better calibration across all calibration metrics. We also see that $\nu$-ensembles achieve significantly higher diversity between ensemble members. These results are consistent across all architectures for both CIFAR-10 and CIFAR-100. For the case of CIFAR-10, we see that the testing accuracy is low, however, this is to be expected due to the small size of the training dataset $Z_{\mathrm{train}}$.

We also compare with Masegosa ensembles (Masegosa, 2020) and Agree to Disagree ensembles (Matteo et al., 2023) (we also attempted to implement DICE ensembles (Ramé & Cord, 2021) but could not replicate a version that converged consistently, despite correspondence with the authors).

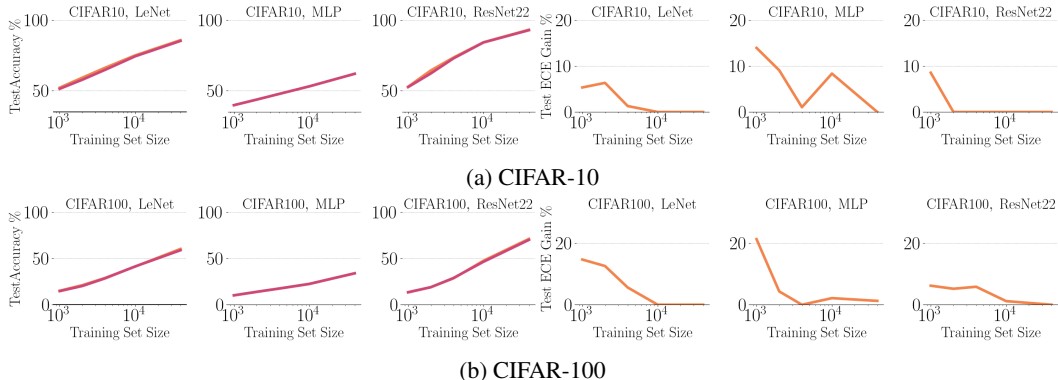

(a) CIFAR-10

(b) CIFAR-100

Figure 2: **Varying the size of the training set.** For both standard and $\nu$-ensembles, we vary the size of the training set $Z_{\text{train}}$ to take values in $\{1000, 2000, 4000, 10000, 40000\}$. $\nu$-ensembles have the same test accuracy as standard ensembles for all training set sizes. We also report the improvement in Expected Calibration Error (ECE) compared to standard ensembles. We see that, as the training size increases, the improvements decrease. Notably, we obtained larger improvements for the more difficult CIFAR-100 dataset than for the easier CIFAR-10 dataset. Also, we continue to have improvements for larger training set sizes. In particular, we observe improvements for the ResNet22 architecture at 10000 training samples while this is not the case for CIFAR-10.

We see that both Masegosa and Agree to Disagree ensembles tend to underfit the data and have worse testing accuracy than $\nu$-ensembles. In particular, Agree to Disagree ensembles also have in general worse calibration. Masegosa ensembles on the other hand have somewhat better calibration than $\nu$-ensembles in most cases. Our algorithm compares very favorably in terms of time and space complexity with both Masegosa and Agree to Disagree Ensembles. Standard and $\nu$ ensembles have $\mathcal{O}(1)$ memory cost as the ensemble size increases, if ensemble members are trained sequentially. On the other hand, Masegosa and Agree to Disagree ensembles in general scale like $\mathcal{O}(K)$ as all the ensemble members have to be trained jointly. Analyzing the computational cost is more complicated, however in general Masegosa ensembles require approximately $\times 2$ the computational time of Standard ensembles. Agree to Disagree ensembles scale roughly as $\mathcal{O}(K)$ as ensemble members have to be computed one at a time. In Figure 4 we compare the computational cost of Standard, $\nu$ and Agree to Disagree Ensembles.

We then explore the effect of increasing the dataset size. We plot the results of varying the training set size in $\{1000, 2000, 4000, 10000, 40000\}$ in Figure 2. We observe that $\nu$-ensembles continue achieving the same accuracy as standard ensembles for all training set sizes. At the same time, they retain large improvements in calibration, in terms of the ECE, for small to medium size training sets. For larger training sets the improvements gradually decrease. Notably, there are differences between the easier CIFAR-10 and the more difficult CIFAR-100 dataset. Our calibration gains are significantly larger for the more difficult CIFAR-100 dataset. Furthermore, we retain these gains for larger training set sizes. In particular, we observe improvements for the ResNet22 architecture and 10000 training samples, while this is not the case for CIFAR-10.

## 5.2 Out-of-distribution (OOD) generalization

We evaluated $\nu$-ensembles and standard ensembles on difficult out-of-distribution tasks for the CIFAR-10 dataset, for the case of 1000 training samples. Specifically, we followed the approach introduced in Hendrycks & Dietterich (2018) which proposed to evaluate the robustness of image classification algorithms to 15 common corruption types. We apply the corruption in 5 levels of increasing severity and evaluate the average test accuracy and calibration in terms of ECE across all corruption types. We plot the results in Figure 3. We observe that $\nu$-ensembles retain the same testing accuracy as standard ensembles. At the same time, they are significantly better calibrated in terms of the Expected Calibration Error. This holds for all tested architectures and for all corruption levels. We note that in the ResNet22 case, we see that $\nu$-ensembles are particularly useful for high-intensity corruptions (the improvement in ECE increases from 10% to 15%).

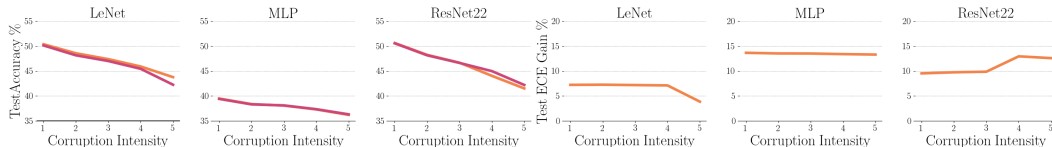

Figure 3: **CIFAR-10 robustness to common corruptions.** We apply 15 common image corruptions to the CIFAR-10 testing dataset for 5 levels of increasing intensity. For each intensity level, we then estimate the average testing accuracy and ECE across all corruption types, for both the standard ensemble and the $\nu$-ensemble. We observe that the $\nu$-ensemble retains approximately the same testing accuracy as the standard ensemble for all corruption levels. At the same time, the $\nu$-ensemble is significantly better calibrated than the standard ensemble.

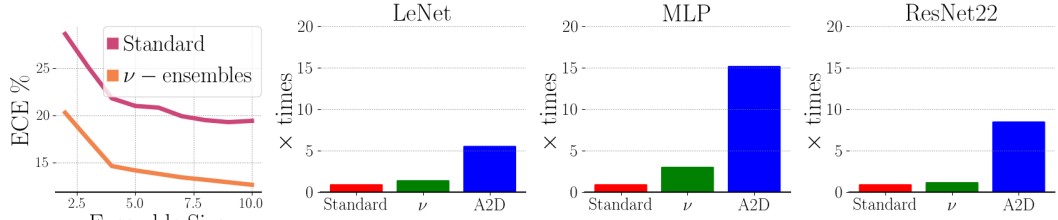

Figure 4: $\nu$**-ensembles and other methods.** Left: Improvements in ECE plateau around 8 ensemble members for Standard ensembles, but continue improving for $\nu$-ensembles. Other figures: we compare the training time of Standard, $\nu$ and Agree to Disagree ensembles, for the CIFAR-10 dataset with 1000 training samples and 5000 unlabeled samples. We plot (total training time)/(epochs ∗ ensemble size). Not only do Agree to Disagree ensembles have to be trained sequentially but the computational complexity for each member is significantly larger.

## 6 Limitations

In our experiments, $\nu$-ensembles demonstrate enhanced calibration performance when applied to standard ensembles, particularly in low to medium-data scenarios. However, in the context of a large data regime, we did not observe any notable improvements. Attempting to force the ensemble to learn random labels in such cases actually had a detrimental effect on calibration. This complex behaviour warrants a more nuanced theoretical analysis. The ability to predict in advance the specific training and unlabeled dataset sizes that would benefit from $\nu$-ensembles would be a valuable asset. Additionally, it is worth noting that despite observing significant enhancements in calibration, counterintuitively we did not observe corresponding improvements in accuracy.

## 7 Conclusion

Deep ensembles have established themselves as a very strong baseline that is challenging to surpass. Not only do they consistently yield improvements across diverse settings, but they also do so with a very simple and efficient algorithm. Consequently, any algorithms aiming to enhance deep ensembles should prioritize efficiency and conceptual simplicity to ensure widespread adoption. In this work, we introduced $\nu$-ensembles, a novel deep ensemble algorithm that achieves both goals. When presented with an unlabeled dataset, $\nu$-ensembles generate distinct labelings for each ensemble member and subsequently fit both the training data and the randomly labeled data. Contrary to other diversity promoring algorithms, $\nu$-ensembles are trivial to parallelize or estimate in a distributed fashion, as each ensemble member can be trained independently of the rest, apart from the final validation step. Future directions of research include exploring the potential for $\nu$-ensembles to outperform standard ensembles in the context of large datasets.

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

# A  Proofs

## A.1  Proof of Theorem 1

**Theorem 2.** *(Theorem 2, Masegosa (2020)) For any distribution $\hat{\rho}$ on $\mathcal{F}$*

$$\mathbf{E}_{(y,\boldsymbol{x})\sim\mathcal{D}}\left[-\ln\mathbf{E}_{\mathbf{w}\sim\hat{\rho}}\left[p(y|\boldsymbol{x},f(\boldsymbol{x};\mathbf{w}))\right]\right] \leq \mathbf{E}_{\mathbf{w}\sim\hat{\rho}}\left[\mathcal{L}_{(y,\boldsymbol{x})\sim\mathcal{D}}^{\ell_{\mathrm{nll}}}(f(\boldsymbol{x};\mathbf{w}))\right] - \mathbf{V}(\hat{\rho}) \qquad (6)$$

*where $\mathbf{V}(\hat{\rho})$ is a variance term defined as*

$$\mathbf{V}(\hat{\rho}) = \mathbf{E}_{(y,\boldsymbol{x})\sim\mathcal{D}}\left[\frac{1}{2\max_{\mathbf{w}}p(y|\boldsymbol{x};\mathbf{w})}\mathbf{E}_{\mathbf{w}\sim\hat{\rho}}\left[(p(y|\boldsymbol{x},\mathbf{w}) - \mathbf{E}_{\mathbf{w}\sim\hat{\rho}}\left(p(y|\boldsymbol{x},\mathbf{w})\right))^2\right]\right]. \qquad (7)$$

We need to bound $\mathbf{V}(\hat{\rho})$ and $\mathbf{E}_{\mathbf{w}\sim\hat{\rho}}\left[\mathcal{L}_{(y,\boldsymbol{x})\sim\mathcal{D}}^{\ell_{\mathrm{nll}}}(f(\boldsymbol{x};\mathbf{w}))\right]$ using their empirical versions. We will use a labeled training set $Z$ to bound the term $\mathbf{E}_{\mathbf{w}\sim\hat{\rho}}\left[\mathcal{L}_{(y,\boldsymbol{x})\sim\mathcal{D}}^{\ell_{\mathrm{nll}}}(f(\boldsymbol{x};\mathbf{w}))\right]$ and an unlabeled set $U$ to bound $\mathbf{V}(\hat{\rho})$. To bound the terms we will use existing PAC-Bayes bounds. The variance term has to be rewritten in the form $\mathbf{E}_{\mathbf{w}\sim\hat{\rho}}\mathbf{E}_{(y,\boldsymbol{x})\sim\mathcal{D}}\left[L(y,\boldsymbol{x},\mathbf{w})\right]$ in which PAC-Bayes bounds are directly applicable.

Let us assume as in Masegosa (2020) that the model likelihood is bounded:

**Assumption 1.** *Masegosa (2020) There exists a constant $C < \infty$ such that $\forall \boldsymbol{x} \in \mathcal{X}$, $\max_{y,\mathbf{w}}p(y|\boldsymbol{x};\mathbf{w}) \leq C$.*

Note that this assumption holds for the classification setting with $C = 1$. Then the variance can be written as

$$\begin{aligned}
\mathbf{V}(\hat{\rho}) &= \frac{1}{2}\mathbf{E}_{(y,\boldsymbol{x})\sim\mathcal{D}}\left[\mathbf{E}_{\mathbf{w}\sim\hat{\rho}}\left[(p(y|\boldsymbol{x},\mathbf{w}) - \mathbf{E}_{\mathbf{w}\sim\hat{\rho}}\left(p(y|\boldsymbol{x},\mathbf{w})\right))^2\right]\right] \\
&= \frac{1}{2}\mathbf{E}_{(y,\boldsymbol{x})\sim\mathcal{D}}\mathbf{E}_{\mathbf{w}\sim\hat{\rho}}\left[p(y|\boldsymbol{x},\mathbf{w})^2\right] - \frac{1}{2}\mathbf{E}_{(y,\boldsymbol{x})\sim\mathcal{D}}\left[\mathbf{E}_{\mathbf{w}\sim\hat{\rho}}p(y|\boldsymbol{x},\mathbf{w})\right]^2 \\
&= \frac{1}{2}\mathbf{E}_{(y,\boldsymbol{x})\sim\mathcal{D}}\mathbf{E}_{\mathbf{w}\sim\hat{\rho}}\left[p(y|\boldsymbol{x},\mathbf{w})^2\right] - \frac{1}{2}\mathbf{E}_{(y,\boldsymbol{x})\sim\mathcal{D}}\left[\mathbf{E}_{\mathbf{w}\sim\hat{\rho}}p(y|\boldsymbol{x},\mathbf{w})\mathbf{E}_{\mathbf{w}'\sim\hat{\rho}}p(y|\boldsymbol{x},\mathbf{w}')\right] \quad (8) \\
&= \frac{1}{2}\mathbf{E}_{(y,\boldsymbol{x})\sim\mathcal{D}}\mathbf{E}_{\hat{\rho}(\mathbf{w},\mathbf{w}')}\left[p(y|\boldsymbol{x},\mathbf{w})^2 - p(y|\boldsymbol{x},\mathbf{w})p(y|\boldsymbol{x},\mathbf{w}')\right] \\
&= \frac{1}{2}\mathbf{E}_{(y,\boldsymbol{x})\sim\mathcal{D}}\mathbf{E}_{\hat{\rho}(\mathbf{w},\mathbf{w}')}\left[L(y,\boldsymbol{x},\mathbf{w},\mathbf{w}')\right]
\end{aligned}$$

where $L(y,\boldsymbol{x},\mathbf{w},\mathbf{w}') = p(y|\boldsymbol{x},\mathbf{w})^2 - p(y|\boldsymbol{x},\mathbf{w})p(y|\boldsymbol{x},\mathbf{w}')$ and $\hat{\rho}(\mathbf{w},\mathbf{w}') = \hat{\rho}(\mathbf{w})\hat{\rho}(\mathbf{w}')$.

We can then use the following PAC-Bayes theorem to lower bound $\mathbf{V}(\hat{\rho})$ through it's empirical estimate, noting that $L(y,\boldsymbol{x},\mathbf{w},\mathbf{w}') \leq 1$ which is a requirement for this bound.

**Theorem 3.** *(PAC-Bayes-$\lambda$, Thiemann et al. (2017)). For any probability distribution $\pi$ on $\mathcal{F}$ that is independent of $U$ and any $\delta_1 \in (0,1)$, with probability at least $1 - \delta_1$ over a random draw of a sample $U$, for all distributions $\hat{\rho}$ on $\mathcal{F}$ and all $\gamma \in (0,2)$ simultaneously and a bounded loss $L \leq 1$*

$$\mathbf{E}_{\mathbf{w}\sim\hat{\rho}}\mathbf{E}_{(y,\boldsymbol{x})\sim\mathcal{D}}\left[L(y,\boldsymbol{x},\mathbf{w})\right] \geq \left(1 - \frac{\gamma}{2}\right)\mathbf{E}_{\mathbf{w}\sim\hat{\rho}}\frac{1}{m}\sum_{(y,\boldsymbol{x})\in U}\left[L(y,\boldsymbol{x},\mathbf{w})\right] - \frac{\mathrm{KL}(\hat{\rho}||\pi) + \ln(2\sqrt{m}/\delta)}{\gamma m} \tag{9}$$

We then turn to the term $\mathbf{E}_{\mathbf{w}\sim\hat{\rho}}\left[\mathcal{L}_{(y,\boldsymbol{x})\sim\mathcal{D}}^{\ell_{\mathrm{nll}}}(f(\boldsymbol{x};\mathbf{w}))\right]$ where $L$ is unbounded due to the NLL loss. We will use the following bound:

**Theorem 4.** *( Alquier et al. (2016)). For any probability distribution $\pi$ on $\mathcal{F}$ that is independent of $Z$ and any $\delta_2 \in (0,1)$, with probability at least $1 - \delta_2$ over a random draw of a sample $Z$, for all distributions $\hat{\rho}$ on $\mathcal{F}$ and $\gamma > 0$*

$$\mathbf{E}_{\mathbf{w}\sim\hat{\rho}}\left[\mathcal{L}_{(y,\boldsymbol{x})\sim\mathcal{D}}^{\ell_{\mathrm{nll}}}(f(\boldsymbol{x};\mathbf{w}))\right] \leq \mathbf{E}_{\mathbf{w}\sim\hat{\rho}}\left[\hat{\mathcal{L}}_Z^{\ell_{\mathrm{nll}}}(f(\boldsymbol{x};\mathbf{w}))\right] + \frac{\mathrm{KL}(\hat{\rho}||\pi) + \ln(\frac{1}{\delta}) + \psi_{\pi,\mathcal{D}}(\gamma,n)}{\gamma n} \quad (10)$$

*where*

$$\psi_{\pi,\mathcal{D}}(\gamma,n) = \ln\mathbf{E}_\pi\mathbf{E}_\mathcal{D}\left[e^{\gamma n\left(\mathcal{L}_{(y,\boldsymbol{x})\sim\mathcal{D}}^{\ell_{\mathrm{nll}}}(f(\boldsymbol{x};\mathbf{w})) - \hat{\mathcal{L}}_Z^{\ell_{\mathrm{nll}}}(f(\boldsymbol{x};\mathbf{w}))\right)}\right]. \tag{11}$$

By setting $\gamma_1 = \gamma_2 = \gamma/2$ and taking a union bound we then get:

**Theorem 5.** *For any probability distribution $\pi$ on $\mathcal{F}$ that is independent of $U$ and $Z$ and any $\delta \in (0,1)$, with probability at least $1 - \delta$ over a random draw of a sample $U$ and $Z$, for all distributions $\hat{\rho}$ on $\mathcal{F}$ and all $\gamma \in (0,2)$ simultaneously*

$$\mathbf{E}_{(y,\boldsymbol{x})\sim\mathcal{D}}\left[-\ln\mathbf{E}_{\mathbf{w}\sim\hat{\rho}}\left[p(y|\boldsymbol{x},f(\boldsymbol{x};\mathbf{w}))\right]\right] \leq$$
$$\mathbf{E}_{\mathbf{w}\sim\hat{\rho}}\left[\hat{\mathcal{L}}_Z^{\ell_{\mathrm{nll}}}(f(\boldsymbol{x};\mathbf{w}))\right] + \frac{\mathrm{KL}(\hat{\rho}||\pi) + \ln(1/\delta) + \psi_{\pi,\mathcal{D}}(\gamma,n)}{\gamma n} \tag{12}$$
$$- \left(1 - \frac{\gamma}{2}\right)\hat{\mathbf{V}}(\hat{\rho}) + \frac{\mathrm{KL}(\hat{\rho}||\pi) + \ln(2\sqrt{m}/\delta)}{\gamma m}.$$

What remains is to define the prior $\pi$ and posterior $\hat{\rho}$ distributions appropriately. We first set $\hat{\rho}(\mathbf{w}) = \frac{1}{K}\sum_i \delta(\mathbf{w} = \hat{\mathbf{w}}_i)$ which denotes an ensemble. We then follow [Masegosa (2020)](#) in properly defining the KL between $\hat{\rho}(\mathbf{w})$ and a given prior. Specifically, we restrict ourselves to a new family of priors, denoted $\pi_F(\mathbf{w})$. For any prior $\pi_F(\mathbf{w})$ within this family, its support is contained in $\mathbf{w}_F$, which denotes the space of real number vectors of dimension M that can be represented under a finite-precision scheme using F bits to encode each element of the vector. So we have $supp(\pi_F) \subseteq \mathbf{w}_F \subseteq \mathcal{R}^M$. This prior distribution $\pi_F$ can be expressed as, $\pi_F(\mathbf{w}) = \sum_{\mathbf{w}'\in\mathbf{w}_F} w_{\mathbf{w}'}\delta(\mathbf{w} = \mathbf{w}')$ where $w_{\mathbf{w}'}$ are positive scalar values parametrizing this prior distribution. They satisfy $w_{\mathbf{w}'} \geq 0$ and $\sum w_{\mathbf{w}'} = 1$. In this way, we can define a finite-precision counterpart to the Gaussian distribution where $w_{\mathbf{w}'} = \frac{1}{A}e^{-||\mathbf{w}'||_2^2}$ and $A$ is an appropriate normalization constant.

Puting everything back in (12) we get

$$\mathbf{E}_{(y,\boldsymbol{x})\sim\mathcal{D}}\left[-\ln\frac{1}{K}\sum_i\left[p(y|\boldsymbol{x},f(\boldsymbol{x};\hat{\mathbf{w}}_i))\right]\right]$$
$$\leq \frac{1}{K}\sum_i\left[\hat{\mathcal{L}}_Z^{\ell_{\mathrm{nll}}}(f(\boldsymbol{x};\hat{\mathbf{w}}_i))\right] - \left(1 - \frac{\gamma}{2}\right)\hat{\mathbf{V}}(\hat{\rho}) + \frac{1}{K}\sum_i h\left(\|\hat{\mathbf{w}}_i\|_2^2\right), \tag{13}$$

where

$$h\left(\|\hat{\mathbf{w}}_i\|_2^2\right) = \frac{\|\hat{\mathbf{w}}_i\|_2^2 + \ln A + K\ln(1/\delta) + K\psi_{\pi,\mathcal{D}}(\gamma,n)}{\gamma n} + \frac{\|\hat{\mathbf{w}}_i\|_2^2 + \ln A + K\ln(2\sqrt{m}/\delta)}{\gamma m}, \tag{14}$$

and which holds for any $\delta \in (0,1)$, with probability at least $1 - \delta$ over a random draw of a sample $U$ and $Z$.

Some further technical points need to be discussed at this point. Formally, Theorem 4 holds for a single value of $\gamma$. In order to combine both PAC-Bayes bounds we would need to form a grid over $\gamma$ in the range $(0,2)$ and do a union bound over this grid. The combined bound would then hold only for values on this grid. This results analysis only results in a negligible loosening of the bound ([Dziugaite & Roy](#), 2017) and as such we neglect this discussion.

Since we have defined our bound in the discrete setting we cannot technically take derivatives of the resulting objective. However, as discussed in [Masegosa (2020)](#) during optimization we simply use the continuous version of all functions, knowing that we will arrive withing a solution of finite precision.

## B Additional conditions for a high-probability bound

Given inequality 13, we can take the expectation over the proposed algorithm, $\hat{\rho} \sim \mathcal{A}$, to obtain

$$\mathbf{E}_{\hat{\rho}\sim\mathcal{A}}\mathbf{E}_{(y,\boldsymbol{x})\sim\mathcal{D}}\left[-\ln\frac{1}{K}\sum_i\left[p(y|\boldsymbol{x},f(\boldsymbol{x};\hat{\mathbf{w}}_i))\right]\right]$$
$$\leq \mathbf{E}_{\hat{\rho}\sim\mathcal{A}}\left[\frac{1}{K}\sum_i\left[\hat{\mathcal{L}}_Z^{\ell_{\mathrm{nll}}}(f(\boldsymbol{x};\hat{\mathbf{w}}_i))\right]\right] - \left(1 - \frac{\gamma}{2}\right)\frac{K-1}{2cK} + \mathbf{E}_{\hat{\rho}\sim\mathcal{A}}\left[\frac{1}{K}\sum_i h\left(\|\hat{\mathbf{w}}_i\|_2^2\right)\right], \tag{15}$$

which holds for any $\delta \in (0,1)$, with probability at least $1 - \delta$ over a random draw of a sample $U$ and $Z$.

Then, setting $L_1(\hat{\rho}) = \frac{1}{K}\sum_i \left[\hat{\mathcal{L}}_Z^{\ell_{\mathrm{nll}}}(f(\boldsymbol{x};\hat{\mathbf{w}}_i))\right]$ and $L_2(\hat{\rho}) = \frac{1}{K}\sum_i h\left(\|\hat{\mathbf{w}}_i\|_2^2\right)$ we note that both $L_1$ and $L_2$ are in general unbounded. To obtain a high-probability bound on $\mathbf{E}_{\hat{\rho}\sim\mathcal{A}}\mathbf{E}_{(y,\boldsymbol{x})\sim\mathcal{D}}\left[-\ln\frac{1}{K}\sum_i [p(y|\boldsymbol{x}, f(\boldsymbol{x};\hat{\mathbf{w}}_i))]\right]$ we need additional conditions on $\mathcal{A}$ namely that it outputs $\hat{\rho}$ such that $L_1(\hat{\rho}) \leq B$ and $L_2(\hat{\rho}) \leq C$ where $B, C$ are positive constants.

Then, for a finite sample $R \in \mathcal{A}^r$ and using Hoeffding's inequality and applying a union bound we can write

$$
\begin{aligned}
\mathbf{E}_{\hat{\rho}\sim\mathcal{A}}\mathbf{E}_{(y,\boldsymbol{x})\sim\mathcal{D}}&\left[-\ln\frac{1}{K}\sum_{i\in\hat{\rho}}[p(y|\boldsymbol{x}, f(\boldsymbol{x};\hat{\mathbf{w}}_i))]\right] \\
&\leq \frac{1}{r}\sum_{\hat{\rho}\in R}\left[\frac{1}{K}\sum_{i\in\hat{\rho}}\left[\hat{\mathcal{L}}_Z^{\ell_{\mathrm{nll}}}(f(\boldsymbol{x};\hat{\mathbf{w}}_i))\right]\right] + \sqrt{\frac{B^2\ln 1/b}{2r}} \\
&\quad -\left(1-\frac{\gamma}{2}\right)\frac{K-1}{2cK} + \frac{1}{r}\sum_{\hat{\rho}\in R}\left[\frac{1}{K}\sum_{i\in\hat{\rho}}h\left(\|\hat{\mathbf{w}}_i\|_2^2\right)\right] + \sqrt{\frac{C^2\ln 1/c}{2r}},
\end{aligned}
$$

which holds with probability $1 - (\delta + b + c)$ over the random draws of $U \in \mathcal{D}^m$, $Z \in \mathcal{D}^n$ and $R \in \mathcal{A}^r$ for $b, c \in (0,1)$. The bound still holds *for the expectation* over $\hat{\rho} \sim \mathcal{A}$ and not with high probability for a single draw from $\mathcal{A}$. It guarantees that on average, ensembles that fit the training data and the randomly labeled data well, while having low complexity will generalize well to unseen data. In our experimental section, however, we have found that optimizing a single ensemble using our $\nu$-ensemble objective achieves all the desirable properties.

## C    Experimental setup

We ran all experiments using A100, and V100 NVIDIA GPUs on our cluster. In total, the experiments consumed approximately 10000 hours of GPU time. The implementations were done in JAX Bradbury et al. (2018). While data loading was done in Tensorflow Abadi et al. (2015). For $\nu$-ensembles, for the LeNet architecture we investigated epochs in the range $[100, 120, 140, 160, 180, 200, 220, 240, 260]$, for the MLP $[100, 120, 140, 160, 180, 200, 220, 240, 260]$, for the ResNet $[200, 220, 250, 270, 300, 320, 350, 370, 400]$. For the regularization strength, we searched in the range $[1, 0.1, 0.05, 0.01, 0]$ and for the optimizer learning rate in $[0.0001, 0.001]$. We investigated the same epoch and learning rate ranges for Standard ensembles. Agree to Disagree ensembles contain a single hyperparameter $\alpha$. We tested values in the range $[1, 0.1, 0.01, 0.001, 0.0001]$.

