# OpenReview forum: "Something for (almost) nothing: improving deep ensemble calibration using unlabeled data"
_NeurIPS.cc/2023/Workshop/WANT — WANT@NeurIPS 2023 Poster_

### Official Review · Reviewer_eoNp · 2023-10-23
**ν-Ensembles: Boosting Neural Network Calibration and Diversity with Random Labels**

**Confidence:** 3

**Review:**

Deep ensembles enhance the accuracy and calibration of neural networks, with their performance tied to the diversity of their member models. This paper introduces ν-ensembles, an algorithm that improves deep ensemble calibration and diversity without increasing computational costs, by generating random labels for unlabeled data and training each ensemble member on these random labels. Through PAC-Bayesian analysis and experiments, the authors show that ν-ensembles outperform standard ensembles in calibration metrics, both for in-distribution and out-of-distribution settings.

Deep networks can perfectly fit random labels. This peculiar property is utilized in ν-ensembles. By fitting to random labels, the members of the ensemble are forced to diversify since each member receives a different random labeling of the same data. This boosts the diversity of the ensemble's predictions, potentially enhancing its robustness and generalization.

Strengths:

In-Distribution Performance:

Calibration: ν-ensembles achieve significantly better calibration in all calibration metrics than standard ensembles. Calibration is an important quality for machine learning models that measures how well the predicted probabilities of a classifier reflect the true outcomes.
The Mutual Information (MI) of ν-ensembles is significantly lower than standard ensembles. This suggests that ν-ensembles are more diverse.
Consistency: The improvements in calibration and diversity are consistent across various architectures and datasets, such as CIFAR-10 and CIFAR-100.

Out-of-Distribution Performance:

ν-ensembles achieve similar test accuracy to standard ensembles but are significantly better calibrated regarding the Expected Calibration Error (ECE) across various corruption levels. This shows robustness to common image corruptions, especially at high-intensity corruptions.

Scalability and Computational Efficiency:

Memory Cost: ν-ensembles have O(1) memory cost (as standard ensembles), eliminating the joint training.
Training Time: ν-ensembles have a competitive computational cost compared to ensemble methods like Agree to Disagree ensembles.

Limitations:

Ν-ensembles have comparable accuracy to standard ensembles, but they do not significantly outperform them.
Decreased Improvements with Larger Training Sets: for larger training sets, the calibration improvements provided by ν-ensembles gradually decrease.
An analysis including more complex datasets would be beneficial to corroborate further the observations reported in Fig. 2.

---

### Official Review · Reviewer_YiKt · 2023-10-24
**deep ensemble method with limited computational and memory requirements**

**Confidence:** 3

**Review:**

The authors introduce a deep ensemble method with the aim of maintaining computational and memory requirements with standard deep ensemble method with PAC-Bayes bound analysis. The simple approach combines the training of a standard deep ensemble with training on an unlabelled datasets where class labels are randomly assigned. A key feature is that the members of the ensemble can be trained in parallel or on a distributed platform. As a result, the proposed method is more efficient than other deep ensemble methods, such as Masegosa and agree to disagree ensemble. The author compare the proposed method to standard deep ensemble, Masegosa and agree to disagree ensemble on CIFAR 10 and 100 in terms of accuracy, ECE, memory and computations. The authors find that while being more efficient, the propose method achieves similar accuracy, often with better calibration on small labeled training sets. The choice of datasets is a limitation as they are small and very similar. Nonetheless, the paper is of good quality and fits well within the scope of the worksop. The manuscript could be improved by discussing the influence of the size of unlabelled dataset, noisy, equal class representation in the unlabelled data and/or out-of-distribution unlabeled data. Some small points, the significance of the colours in Table 1 are missing as is a legend in Figure 2.

---

### Official Review · Reviewer_KCu1 · 2023-10-25
**An interesting idea and well-written paper, though focussed on small-scale problems**

**Confidence:** 3

**Review:**

# Summary
The paper presents a new algorithm (v-ensembles) for encouraging diversity among the members of deep ensembles which, unlike existing work in this area, still allows the ensemble methods to be trained in parallel. The authors consider a classification setting where some labelled and some unlabelled training data is available. They label the unlabelled data at random seperately for each member of the ensemble, ensuring that each data point recieves a different class label for each member of the ensemble, and then train on both the labelled and pseudo-labelled data. The objective is derived from a PAC-Bayes bound. They demonstrate their algorithm on CIFAR-10/100, showing that, when relatively little labelled data is available, they achieve better in and out-of distribution calibration while maintaining accuracy.

# Overall opinion
I suggest: accept if deemed relevant.

I'm not familiar enough with the literature to comment on the novelty, nor can I judge the PAC-Bayes bound.

This is a promising idea that would be interesting to discuss at a workshop. The relevance to this workshop is that the algorithm allows the ensemble members to be trained independently in parallel, which is not possible for other diversity encouraging methods. However, the paper admits that the advantage of the method appears to mainly be for datasets of a few thousand training points or less, in which case we can probably train a model in a few minutes on a single GPU. So perhaps it's not that relevant this particular workshop, but I defer to the organisers. It is definitely an interesting initial work!

# Strengths
- Very clearly written
- Simple idea that's practical to implement
- Promising results when we have a few thousand labelled traning points

# Points that could be improved
- Is the unlabelled data also used by the other methods (e.g. the standard ensemble)? If yes, does this hurt performance over just training on the labelled data?
- The other diversity methods are only included in the comparison of 1000 points, while v-ensembles and standard ensembles are compared up to 40,000 points. Given that v-ensembles stop offering a benefit after 10,000 points, it would have been good to see how the other methods do with more data.
- Is it useful to include results for MLPs on CIFAR? Do people use MLPs for this in practice?
- I suggest bolding the best method in results tables, rather than v-ensembles every time

---

### Meta-Review · Area_Chair_NxTx · 2023-10-27

**Recommendation:** Accept (Oral)
**Confidence:** 4

**Metareview:**

All the reviewers are essentially in a favor of acceptance, I concur. I reccommend the authors to take reviewer comments into account for future submission.

---

### Decision · Program_Chairs · 2023-10-28

**Decision:**

Accept (Poster)

**Comment:**

We thank the authors for their time and contribution to WANT and we are pleased to share that after the reviewing process the paper has been accepted. Congratulations! We encourage the authors to consider reviewers' feedback for the improvement of the camera-ready version. We hope to see you in person at the workshop and brainstorm on efficient training research together!